# Insular Cell Integrity Markers Linked to Weight Concern in Anorexia Nervosa—An MR-Spectroscopy Study

**DOI:** 10.3390/jcm9051292

**Published:** 2020-04-30

**Authors:** Simon Maier, Kathrin Nickel, Evgeniy Perlov, Alina Kukies, Almut Zeeck, Ludger Tebartz van Elst, Dominique Endres, Derek Spieler, Lukas Holovics, Armin Hartmann, Michael Dacko, Thomas Lange, Andreas Joos

**Affiliations:** 1Department of Psychosomatic Medicine and Psychotherapy, Medical Center—University of Freiburg, Faculty of Medicine, University of Freiburg, 79104 Freiburg, Germany; alinakukies@hotmail.com (A.K.); almut.zeeck@uniklinik-freiburg.de (A.Z.); derek.spieler@uniklinik-freiburg.de (D.S.); lukas.holovics@uniklinik-freiburg.de (L.H.); armin.hartmann@uniklinik-freiburg.de (A.H.); andreas.joos@uniklinik-freiburg.de (A.J.); 2Department of Psychiatry and Psychotherapy, Medical Center—University of Freiburg, Faculty of Medicine, University of Freiburg, 79104 Freiburg, Germany; kathrin.nickel@uniklinik-freiburg.de (K.N.); evgeniy.perlov@lups.ch (E.P.); tebartzvanelst@uniklinik-freiburg.de (L.T.v.E.); dominique.endres@uniklinik-freiburg.de (D.E.); 3Luzerner Psychiatrie, Hospital St. Urban, 4915 St. Urban, Switzerland; 4Department of Radiology, Medical Physics, Medical Center—University of Freiburg, Faculty of Medicine, University of Freiburg, 79106 Freiburg, Germany; michael.dacko@uniklinik-freiburg.de (M.D.); thomas.lange@uniklinik-freiburg.de (T.L.); 5Department of Psychosomatic Medicine and Psychotherapy, Ortenau Klinikum, Teaching Hospital of University of Freiburg, 77654 Offenburg, Germany

**Keywords:** anorexia nervosa, magnetic resonance spectroscopy, MRS, insula, glutamate, *N*-acetylaspartate, NAA

## Abstract

Objective: An insular involvement in the pathogenesis of anorexia nervosa (AN) has been suggested in many structural and functional neuroimaging studies. This magnetic resonance spectroscopy (MRS) study is the first to investigate metabolic signals in the anterior insular cortex in patients with AN and recovered individuals (REC). Method: The MR spectra of 32 adult women with AN, 21 REC subjects and 33 healthy controls (HC) were quantified for absolute *N*-acetylaspartate (NAA), glutamate + glutamine (Glx), total choline, myo-inositol, creatine concentrations (mM/L). After adjusting the metabolite concentrations for age and partial gray/white matter volume, group differences were tested using one-way multivariate analyses of variance (MANOVA). Post-hoc analyses of variance were applied to identify those metabolites that showed significant group effects. Correlations were tested for associations with psychometric measures (eating disorder examination), duration of illness, and body mass index. Results: The MANOVA exhibited a significant group effect. The NAA signal was reduced in the AN group compared to the HC group. The REC and the HC groups did not differ in metabolite concentrations. In the AN group, lower NAA and Glx signals were related to increased weight concern. Discussion: We interpret the decreased NAA availability in the anterior insula as a signal of impaired neuronal integrity or density. The association of weight concern, which is a core feature of AN, with decreased NAA and Glx indicates that disturbances of glutamatergic neurotransmission might be related to core psychopathology in AN. The absence of significant metabolic differences between the REC and HC subjects suggests that metabolic alterations in AN represent a state rather than a trait phenomenon.

## 1. Introduction

Patients with anorexia nervosa (AN) persistently restricted their food intake because of severe weight concerns and body image disturbance. AN mainly affects young women [1,2,3] and has the highest mortality rate of all mental disorders [4,5], while only about half of such patients fully recover [1,2].

Two decades ago, the first magnetic resonance spectroscopy (MRS) studies searched for metabolic anomalies associated with AN. MRS is a non-invasive and nonradioactive method for the in vivo detection of several neurometabolites. MRS sequences that measure the ^1^H chemical shift allow the simultaneous absolute quantification of the concentrations of *N*-acetylaspartate (NAA) + *N*-acetyl-aspartyl-glutamate (NAAG), glutamate (Glu) + glutamine (Gln), glycerophosphorylcholine + phosphocholine (total choline, t-Cho), myo-inositol (mI), phosphocreatine (PCre), and creatine (Cre).

NAA is relatively abundant in healthy neuronal tissue and is therefore often used as a marker for neuronal integrity and density [6]. However, it has also been detected in oligodendrocytes and myelin [7,8], and it is synthesized in the mitochondria. Decreased concentrations of NAA indicate a loss of neuronal cells. Although NAA signal differences have been reported in various studies on AN, the results were inconsistent regarding a diminished or elevated NAA level [9,10,11,12,13] (see Table 1).

Glu is the primary excitatory neurotransmitter in the human brain (~85% of all synapses in the central nervous system are glutamatergic) [14] and has been implicated in different psychiatric disorders associated with AN, including anxiety disorders, depression, and obsessive-compulsive disorder [15,16,17,18]. Elevated Glu has toxic effects on neurons as well as oligodendrocytes, astrocytes, and endothelial and immune cells [19]. The distinction of Glu from its precursor and the metabolite Gln is difficult because of a considerable overlap of the spectral peaks. Therefore, the combined Glu and Gln resonances are often denoted collectively as Glx [20]. Previous MRS studies have reported lower Glx [10,12,21] and Glu [22] in the AN group compared to HC subjects or in response to AN-related symptoms; while this has mainly been in frontal cortices, it has also been in the basal ganglia and occipital lobe [22]. Notably, one study reported higher Glx signals in AN [9].

The major part of the t-Cho signal results from glycerophosphorylcholine and phosphocholine, but it also comes from other choline-containing compounds. Glycerophosphorylcholine and phosphocholine are integral components of the cell membrane and are more highly concentrated in glial cells than they are in neurons [23]. Changes in the t-Cho signal have been associated with changes in cell proliferation or cell degeneration [24]. MRS studies that detected t-Cho differences in AN or in relation to anorectic symptoms reported increased concentrations [9,11,12,13,25]. 

Creatine and phosphocreatine are measured as combined resonances (Cre) and are involved in the creatine kinase cycle. Despite their important role in energy metabolism, the level of the Cre signal remains relatively constant, even under conditions of increased energy demand [26]. Therefore, the Cre signal is widely used as an internal reference in MRS studies. While MRS studies reporting Cre signal differences in AN mostly showed increased Cre [9,12,25] in the anterior cingulate and frontal cortices, lower Cre was observed in the dorsolateral prefrontal cortex (dlPFC) [21].

The concentration of the sugar mI is thought to be increased in glial cells compared to neurons, and it serves as an intracellular second messenger. For this reason, mI is used as a glial marker [27]. In AN, mI was found to be decreased [10,21,22,28] whenever mI differences were detected.

Small sample sizes, low MR field strengths, and differences in voxel positioning are likely to be the main cause of the heterogeneous results of previous ^1^H-MRS studies (Table 1). Given that higher field strengths are particularly recommended for the quantification of Glx, and considering that the Cre signal, which was often used as a reference in earlier studies [9,21,25], often reveals inconsistent results, older studies must be interpreted cautiously.

In conclusion, no metabolite was consistently altered across all studies, but increased t-Cho and Cre and decreased mI were the most consistent findings (Table 1). In general, the results of newer studies with higher field strengths and more advanced MRS protocols in combination with larger sample sizes are more reliable.

A longitudinal ^1^H-MRS study reported a normalization of increased t-Cho/Cre ratios with recovery [29], while a small study of adolescent females with AN also indicated normalization, which was (in this case) decreased NAA in the frontal gray matter (GM) [10].

In terms of region of interest (ROI) localization, the anterior insular cortex appears to be of importance. The relevance of this region in AN has been emphasized in various brain structural [33,34,35,36,37] and functional [38,39,40,41] studies, and it even led to the postulation of the so-called “*insula hypothesis*” of AN [42,43]. This hypothesis states that early developmental damage in combination with socio-cultural and other stressors, such as dieting and hormonal changes, may lead to an impairment of insula function. The insula, as a central hub, conveys the information of numerous cortical and subcortical brain areas and therefore plays a central role in the interoceptional awareness and monitoring of the bodily state (somatic marker hypothesis) [44,45]. This includes, among other functions, the experience of physiological correlates of fear and anxiety [46] as well as taste [47,48], hunger [49], disgust [50], and visceral information [51]. Hence, altered insula functioning could explain the dysfunction of interoceptive awareness in AN in turn by not only biasing the experience of one’s own bodily condition or the pleasantness/valence of consumed food but also the reward-related and motivational aspects of food intake [52]. Despite the high relevance of the insular lobe in AN pathomechanism, this area has never been targeted by previous MRS studies.

### Rationale of our Study

This study examined female adults with AN, female adults who had recovered (REC) from AN, and healthy controls (HC) via MRS, which targeted the insular cortex. Considering the results of earlier studies, we expected to observe increased Cre and t-Cho signals in women with AN and decreased mI, while no clear hypothesis regarding NAA and Glx could be derived from previous studies. All hypothetical differences were expected to normalize in the REC subjects.

## 2. Methods and Materials

### 2.1. Participants

The MR spectra of 35 female patients with AN were acquired. Patients were recruited at the in- and outpatient units of the Department of Psychosomatic Medicine and Psychotherapy at the University of Freiburg as well as cooperating hospitals. Diagnoses were made by experienced and board-certified psychiatrists and psychologists and were confirmed by the structured clinical interviews SCID-I and SCID-II [53,54]. Diagnostic criteria according to the DSM-5 [55] were relevant for study inclusion. All AN patients met the weight criterion of a body mass index (BMI) < 18.5 kg/m^2^. Three patients had to be excluded because of either data loss or corruption. Of the 32 remaining AN patients, 3 were of the binge eating/purging subtype, while all others were of the restrictive subtype.

Twenty-two REC were recruited for MRS acquisition and fulfilled the following recovery criteria [56]: No eating disorder symptomatology for at least 12 months with a conservative eating disorder examination total score (EDE) total score [57,58] within 1 standard deviation of normal. For study inclusion, a minimal BMI of >20 kg/m^2^ was envisaged, which most REC participants met. However, the BMI of four REC subjects was slightly below 20 kg/m^2^ (19.3–19.8 kg/m^2^). These participants were clinically completely recovered and had never exceeded a BMI >20 kg/m^2^. In addition, 2 REC subjects had a BMI of 18.5–19.0 kg/m^2^ with similar features, while 1 REC participant had to be excluded because of data loss or corruption, resulting in a final sample size of 21 individuals for the REC group. Finally, 1 REC participant was of the binge eating/purging subtype, while all others were of the restrictive subtype.

HC subjects were recruited via advertisements in local newspapers and announcements on the notice board of the participating hospital. The MR spectra of 40 age-matched female HC subjects were available. Mental disorders in HC subjects were ruled out by SCID-I and II interviews. After 7 HC participants were excluded because of either data loss or corruption, the final HC sample size was 33 individuals.

The general exclusion criteria included any history of head injury or surgery, neurological disorders, severe psychiatric comorbidities (psychosis, bipolar disorders, substance abuse), current regular psychotropic medication, and an inability to undergo MRI scans (e.g., metal implants, claustrophobia). In total, 7 AN, 17 HC, and 10 REC subjects took hormonal contraceptives. One woman with AN had only just started a minor dose of citalopram (10 mg/day; serum level of 31 ng/mL, which is below the therapeutic range) and was thus considered appropriate to be included.

The participants who took no hormonal contraceptives were studied in the luteal phase of the menstrual cycle. Regarding hormonal contraception, they had to be in the progesterone and estrogen phase (i.e., similar to the physiology of the luteal phase). Given that, the AN participants were mostly amenorrhoeic, the phase of their menstrual cycle could not be assessed.

### 2.2. Ethics Statement

All subjects gave written informed consent prior to participation, and the study was approved by the ethics committee of the University Medical Center Freiburg (Approval ID: EK-Freiburg 520/13 June 2013).

### 2.3. Psychometric Assessment

Apart from the SCID interviews [53,54] and the EDE interviews [57,58] mentioned above, the participants completed self-report questionnaires to assess their eating disorder pathology using the Eating Disorder Inventory–2 [59,60] and to assess any depression using the Beck Depression Inventory-II, BDI-II [61,62,63].

### 2.4. Procedure Before Scanning

Starting between 7:30 a.m. and 8:00 a.m., participants received a standardized breakfast, and their consumed calories were assessed. The participants read study-relevant information and completed an MR-safety questionnaire.

### 2.5. Data Acquisition and Processing

All measurements were performed at the University Medical Center of Freiburg using a 3T MAGNETOM Prisma scanner (Siemens Healthineers, Erlangen, Germany), which was equipped with a 20-channel head coil.

Before the MRS measurement was taken, a T1-weighted MPRAGE sequence (repetition time: 2000 ms, echo time: 4.11 ms, flip angle: 12°, field of view: 256 × 256 mm, 160 slices, voxel size: 1 × 1 × 1 mm) was performed and used for manual localization of spectroscopic voxels in the left insular cortex (size: 20 × 20 × 20 mm). A point resolved spectroscopy (PRESS) sequence with an echo time of 30 ms, a relaxation time of 3000 ms, 256 averages, and with water saturation was used. The water reference spectrum was obtained using 16 averages of the same PRESS sequence without water saturation. Shimming parameters were further manually adjusted to minimize the full-width at half maximum of the water peak. The established linear combination of model spectra (LCModel 6.3, Oakville, ON, Canada) software was used for spectra fitting and quantification [64]. The absolute metabolite concentrations of Cre, NAA, Cho, Glx, and mI were estimated [64,65,66]. All spectra were visually controlled to fulfill the internal criterion of quality (i.e., adequate visible peaks of main metabolites), and only spectra with Cramér-Rao lower bounds (CRLBs) for the main metabolites below 20% were included in further analyses. All voxels were segmented into GM, white matter (WM), and cerebrospinal fluid (CSF) using the statistical parametric mapping software SPM12 (Wellcome Trust Centre of Imaging Neuroscience, London, United Kingdom; for details, see [67]), according to the co-registered voxel position of the corresponding morphological T1-weighted image.

### 2.6. Statistical Analyses

The metabolite concentrations, as assessed in the LCModel, were transferred to a data table together with all clinical and psychometric data. All statistical analyses were performed using R statistical computing (version 3.4, R Foundation for Statistical Computing, Vienna, Austria [68]). All dependent variables of interest (metabolite concentrations) were first linearly adjusted for differences in age and partial GM+WM volume using the “predict” function of the R stats package. The resulting adjusted metabolite concentrations were then tested for normality of distribution using the Shapiro–Wilk test of normality. Subsequently, the Levene’s test for homogeneity of variance across groups was computed for the adjusted metabolite concentrations.

Because the adjusted metabolite concentrations of NAA/NAAG, Cre/PCre and mI were not normally distributed, we tested group differences with a robust one-way MANOVA [69] using the rrcov package (version 1.4–3) in R [70]. In the robust MANOVA, a classical Wilk’s Lambda statistic for testing the equality of the group means was modified into a robust one through substituting the classical estimates by the highly robust and efficiently reweighted minimum covariance determinant estimators for mean and variances. The procedure used 10,000 trials for the simulations to compute the multiplication factor for the approximate distribution of the robust Lambda statistic and the degrees of freedom.

To determine which metabolites significantly differed between groups, a post-hoc heteroscedastic one-way ANOVA for trimmed means was carried out with the WRS2 package [71,72]. A *p*-level of <0.05 was chosen as the criterion for significance. The explanatory power is reported using ξ as a robust measure of effect size [73]. A post-hoc Lincon test (with robustness equivalent to the Tukey-Kramer test) was carried out to assess the direction of the effects [71].

The robust MANOVA was repeated with adjusted metabolite concentrations after additionally correcting for the influence contraceptive use.

Finally, the adjusted metabolite concentrations were correlated with psychometric measures of AN (EDE scores, BMI, and duration of illness) using robust correlations (pbcor function of the WRS2 package [72]) independently for the AN group and across all groups (but not for duration of illness). The resulting *p*-values of the correlation analyses were corrected for multiple comparisons using the Benjamini and Hochberg False Discovery Rate [74].

## 3. Results

### 3.1. Demographic and Psychometric Data

The demographic and psychometric data of the participants are summarized in Table 2. For the final data analysis, 32 patients with AN, 21 REC participants, and 33 HCs were included in the study. The REC group was older than were both the AN and HC participants. As expected, there was a difference with respect to BMI between all three groups, with the lowest BMI in patients with AN (AN < REC < HC). The AN group showed more depressive symptoms in the BDI-II than did the REC and HC participants (HC > REC > AN). Diagnoses in the AN group included seven participants with depression, one with a specific phobia, one with obsessive-compulsive disorder, and two with generalized anxiety disorder. Two participants of the REC group had specific phobias. One participant of the AN, two of the HC, and none of the REC group were left-handed.

### 3.2. MRS

After filtering all CRLBs above 20%, the Cho signal of one HC had to be removed from further analyses.

The robust one-way MANOVA model with the five metabolites as independent variables revealed a significant group effect (Λ = 0.712, χ^2^ (8.41) = 17.1, *p* = 0.036). The robust one-way ANOVAs performed *post-hoc* were significant for the NAA peak (F (2,31.4) = 3.760; *p* = 0.034, ξ = 0.35), with the AN group showing lower NAA concentrations (Table 3, Figure 1). These differences remained significant after applying false discovery rate (FDR) correction for multiple comparisons. When comparing the AN group to the REC group the NAA peak differences reached the trend level, but this result failed to survive FDR correction. Both the REC and HC groups exhibited no significant differences in any metabolites. The robust one-way MANOVA model did still show a significant group effect after adjusting the data for the influence of contraceptive use (Λ = 0.723, χ^2^ (8.41) = 16.3, *p* = 0.047).

Partial GM, WM, and CSF volumes of the left insular MRS voxel exhibited no significant group differences (Table 2).

In the AN group, correlation analyses with dimensional measures of AN severity showed a significant negative association of Glx concentrations with EDE shape concern as well as NAA and Glx concentrations with the EDE weight concern score (Table 4, Figure 2). There were no correlations with either the BMI (Cho: *p* = 0.835, Ins: *p* = 0.361, NAA: *p* = 0.870, Glx: *p* = 0.557, Cre: *p* = 0.664) or the duration of illness (Cho: *p* = 0.397, Ins: *p* = 0.508, NAA: *p* = 0.275, Glx: *p* = 0.016, Cre: *p* = 0.223). Only the correlation of NAA and Glx signals with the EDE weight concern score remained significant after FDR correction.

Additional robust correlations between the five different metabolites showed a significant correlation across the concentration of all metabolites (Appendix A).

## 4. Discussion

This MRS study is the largest to date to investigate insular metabolic alterations in women with acute AN and individuals who had recovered from AN. It was found that AN patients exhibited a significantly lower concentration of NAA in the anterior insular cortex in comparison to the REC and HC subjects, while the concentrations of all other investigated metabolites (t-Cho, mI, Glx, Cre) were unaltered. The REC and HC subjects showed no differences in the concentration of metabolites.

In the AN group, concerns about body weight (as measured with the EDE) were associated with lower NAA and Glx availability in the insular cortex. Prior to correction for multiple comparisons, shape concerns also showed a negative association with Glx concentration. It might be questioned whether this finding is due to a very low BMI or a long illness duration. Neither the BMI nor the duration of illness was associated with insular NAA concentrations.

### 4.1. Comparison to Earlier MRS Studies

A comparison to earlier MRS studies is complicated by the fact that this study is the first to target the anterior insular cortex in AN. Only two earlier studies reported lower NAA signals in WM and frontal GM [10,13], while one reported increased NAA in anorectic women in the anterior cingulate cortex and parieto-occipital WM [9]. Our findings complement previous studies that showed a reduction in the GM and WM volumes and the integrity during the acute stage of AN [35,75,76,77]. The detected decreased NAA signal as a marker of neuronal integrity and density [6] in AN might therefore support these earlier results. Structural changes during acute AN appear to be more pronounced in some cortical areas, including the insula [34,78]. Brooks et al. [34] found reduced GM in the right insular cortex in restrictive AN patients. Although the insular cortex has not been consistently affected across structural studies [35,79], in a study of our group with largely overlapping samples, reduced insular volumes and cortical thinning could be observed [78]. Regionally different concentrations of metabolites could thus, in principle, result from differently pronounced structural changes. To determine whether there are changes in metabolite concentration that go beyond known GM/WM volume changes in AN, one has to correct for partial GM/WM volume effects, which was done in the current study. Not all earlier MRS studies statistically controlled for partial GM and WM volume content of the MRS voxel. Metabolic differences that survive such a correction indicate a metabolic disturbance beyond the direct GM or WM volume effect and might therefore be associated with a loss of neuronal integrity, density, or other metabolic changes, some of which might not be known.

### 4.2. Possible Implications of the Findings: Beyond Density, Volume, and Integrity Effects

Our data indicated that lower NAA concentrations were associated with increased concerns about body weight (according to the EDE), which is a core feature of AN. The Glx signal showed a similar association with weight concerns and both NAA and Glx availability were important for glutamatergic neurotransmission [80,81]. Glx was also associated with shape concern, which did not survive the correction for multiple comparisons. Weight and shape concerns are central constructs of AN psychopathology and are part of a disturbed processing of body signals and integration of one’s own body perception. Weight concern refers to the importance of body weight, dissatisfaction with one’s own body weight, and the desire to lose weight as well as the preoccupation with shape and weight. Shape concerns imply feelings of fatness, fear of weight gain, and discomfort when looking at one’s own body (for further details see: [57,58]). Therefore, the lower the Glx and NAA levels are, the more serious are these concerns.

Importantly, because there were no correlations of NAA and Glx with the BMI and we have corrected for the partial GM and WM volume, the correlation most likely does not simply reflect GM or body weight-related deficits. The associations of insular NAA and Glx availability with weight problems and, in the case of Glx, shape concerns, suggest that a lack of Glx or loss of neuronal density/integrity (which are predominantly glutamatergic in the neocortex) may be related to an exacerbation of AN symptoms, presumably because of a decreased glutamatergic effect.

Disturbed glutamatergic action in the insular cortex might therefore be related to AN’s symptoms of “bodily awareness,” such as perception of one’s own body weight, state, and shape [43,82] and reflect dysregulation within the insular lobe (particularly in anterior parts), which serve to integrate multiple pieces of information. The insular cortex is highly connected and, according to Damasio’s somatic marker hypothesis, the insula plays a central role in integrating various perceptions into a cognitive representational map. These perceptions include the information on the bodily state, the state of the autonomic nervous system, appetite, hunger, and disgust [43,44]. Although these are possible associations, no causal relationships can be deduced from these cross-sectional data (i.e., that cerebral dysfunction causes AN psychopathology). Conversely, aberrant perceptions and behavior in AN might lead to metabolic anomalies. A unifying explanation could be that both initially mild behavioral or perceptual abnormalities and initially minor cerebral aberrations reinforce each other in a kind of vicious circle until a full-blown psychopathology and pathophysiology manifests itself.

The absence of the NAA signal reductions in women who had recovered from AN compared to those with AN suggests that the apparent loss of neuronal density/integrity (apart from volume changes, e.g., due to dehydration effects) seem to be of particular importance in acute patients. This aligns with brain volumetric and other structural studies, which report a normalization of WM and GM changes after recovery [37,75,78,83]. Importantly, these patients also recovered with regard to psychopathology—as did our sample—which therefore does not allow any definite conclusions on causality between metabolic changes and psychopathology. At least in the AN group, a simple association with body weight seems unlikely.

However, we cannot completely rule out that the signal similarities of the REC and HC subjects can be attributed to a REC subgroup with a particularly good forecast; only long-term follow-up studies would be able to clarify this point [84].

A recent functional study on anxiety processing in AN showed marked dysfunction of key areas of the so-called “fear network,” including the anterior insula cortex. [84]. Other studies have shown that functional abnormalities in the insula are not only in acute AN but also in REC subjects [38,39,41,85]. These observations suggest that NAA changes are likely to reflect anomalies of neurochemical processes beyond cell density or volume effects [37,75,78,83]. NAA is synthesized in neuronal mitochondria and finally absorbed and metabolized by glial cells. For example, in a longitudinal study on bipolar depression in which the NAA signal was regarded as a state marker for the disorder, the reduced NAA levels were interpreted as a dysregulation of the glial function [86].

### 4.3. Limitations

A longitudinal study design to test for the reversibility of the metabolic differences within the subjects would have been preferable and is envisaged for a subgroup of the AN patients of this study. Another problem was the presence of GM and WM volume effects after correcting for the volume of CSF in the LCModel. To the best of our knowledge there are no biological models available to adjust the concentrations of the different metabolites according to their abundance in GM or WM tissue, respectively. The overlap of the reported structural, functional, and metabolic alterations in the insular cortex in AN, which was investigated in the same subjects, should be confirmed in an independent sample.

## 5. Summary

In this study, we found decreased NAA signals in the anterior insular cortex of women with AN and lower NAA and Glx concentrations in patients with stronger AN symptoms. These changes cannot be explained by mere GM and WM volume changes, but they could be an expression of reduced neuronal and glial cell density, integrity, or function. These metabolic changes may reflect altered neurotransmission; thus, they may be related to anomalies of insula functioning that are known from functional MRI studies. This possible association should be further investigated in future combined MRS+fMRI studies. The REC and HC participants showed no differences in metabolite concentrations, which is encouraging for both patients and therapists. These data also improve our understanding of the pathophysiology of acute and recovered patients with AN, which can influence both our clinical thinking and psychoeducational interventions [87].

## Figures and Tables

**Figure 1 jcm-09-01292-f001:**
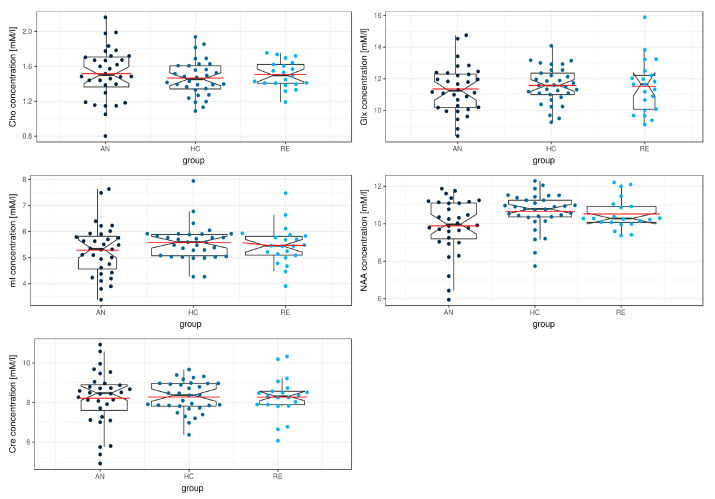
Boxplot of left insula metabolite concentrations across the anorexia nervosa (AN), healthy control (HC) and recovered group (RE). Red lines indicate mean metabolite concentrations.Abbreviations: Cho = choline; mI = myo-inositol; NAA = *N*-acetylaspartate; Glx = glutamate+glutamine; Cre = creatine.

**Figure 2 jcm-09-01292-f002:**
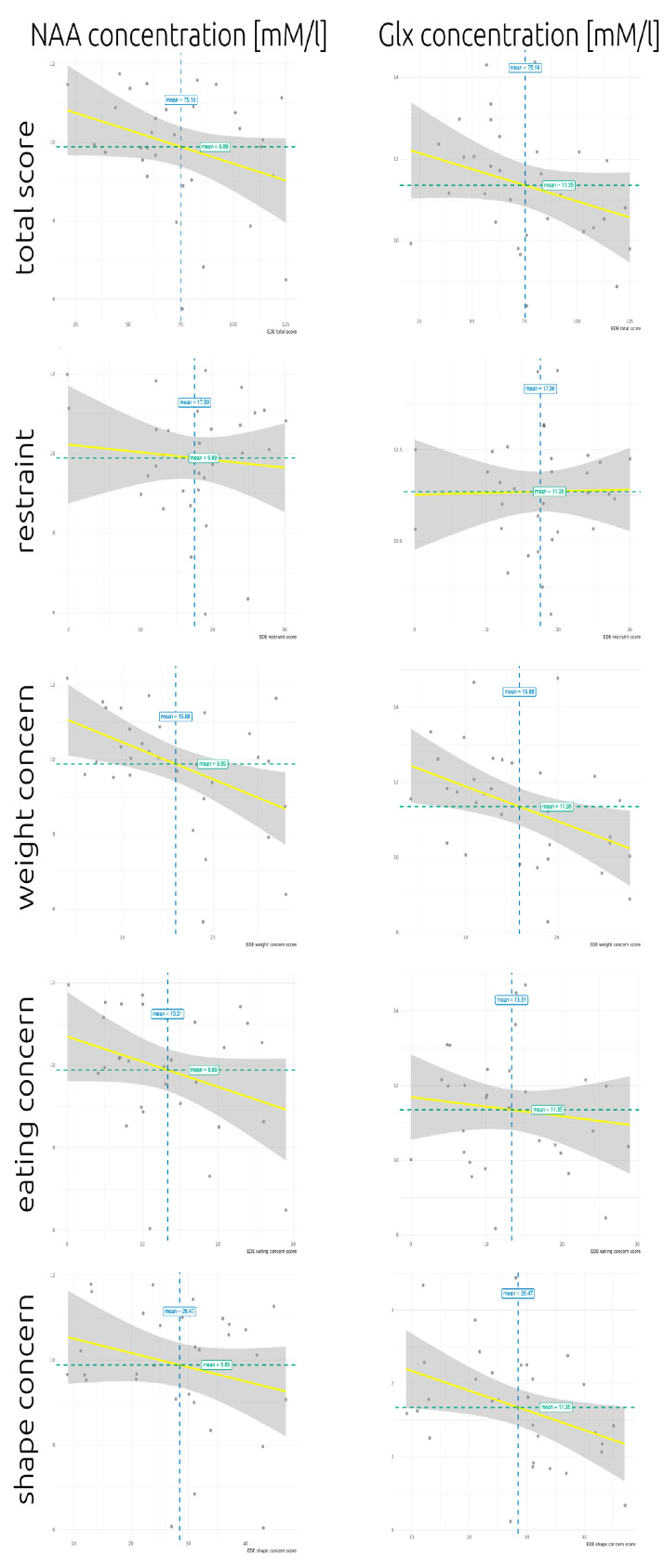
Scatterplot of eating disorder examination (EDE) total scores and subscores of the Anorexia group with left insular *N*-acetylaspartate (NAA) and glutamate + glutamine (Glx) concentrations, respectively. Blue lines indicate mean EDE scores. Green lines indicate mean metabolite concentrations. Regression lines are depicted in yellow with the 95% confidence interval as grey area. Significant correlations are NAA with weight concerns and Glx with weight and shape concerns.

**Table 1 jcm-09-01292-t001:** Overview of previous magnetic resonance spectroscopy (MRS) studies in anorexia nervosa.

Author	Patient Collective	Age	Sex	Patients/Controls	MR Method	Target Region	Results
Kato et al. (1997) [29]	AN (3 comorbid BN)	18–32	female	4 (2 longitudinal)/13	Localized ^31^P (TE = 20 ms) 1.5T	Frontal lobe	PDE/*P* total + decreasing after gaining weight
Schlemmer et al. (1998) [13] *	AN	16 ± 1.9	female	10/17	^1^H SVS STEAM (TE = 50 ms) 1.5T	(1) Thalamus (2) parieto-occipital WM	(1) Cho/Cr+, NAA/Cho−
Möckel et al. (1999) [30] *	AN	15.7 ± 1.7	female	22/17 (11 longitudinal)	^1^H SVS STEAM (TE = 50 ms) 1.5T	(1) Thalamus (2) parieto-occiptal WM	Both: Cho/Cr+ normalization with recovery
Hentschel et al. (1999) [11] *	AN	15.7 ± 1.7	female	15/17	^1^H SVS STEAM (TE = 50 ms) 1.5T	(1) Thalamus (2) parieto-occiptal WM	(1) Cho/Cr+ NAA/Cr+
Roser et al. (1999) [28]	AN and BN	10–28	19 female 1 male	20/15	^1^H SVS STEAM (TE = 20 ms) 1.5T	(1) Frontal white matter (2) occipital gray matter, (3) cerebellum	(1) mI/Cr− lipid/Cr− (2) lipid/Cr− (3) all metabolites + except lipids
Rzanny et al. (2003) [31]	AN	12–20	female	10/10	Localized ^31^P (TE = 16 ms) 1.5T	Frontal lobe	PDE/*P* total−
Ohrmann et al. (2004) [21]	AN	22.7 ± 3.8	female	10/12	^1^H SVS STEAM (TE = 20 ms) 1.5T	(1) Rostral ACC (2) dlPFC	(1) Glx− (2) Cr−, mI−
Grzelak et al. (2005) [32]	AN	16–22	female	10/10	^1^H SVS STEAM (TE = 20 ms) 1.5T	Parietal WM, parietal GM	WM: lipid−
Castro-Fornieles et al. (2007) [10]	AN	11–17	female	12/12	^1^H SVS PRESS (TE = 35 ms) 1.5T	Frontal grey matter	NAA−, Glx−, ml− 7 months follow-up NAA+
Castro-Fornieles et al. (2010) [25]	AN vs. short-term recovered	13–18	female	32/20	^1^H SVS PRESS (TE = 35 ms) 1.5T	prefrontal	Cho+, Cr+
Joos et al. (2011) [12]	10 BN 7AN	24.8 ± 4.9	female	17/14	^1^H SVS PRESS (TE = 30 ms) 3T	ACC	No difference: NAA, Cho, Cr, mI, Glu, Glx
Blasel et al. (2012) [9]	AN	14.4 ± 1.9	female	21/29	^1^H SVS PRESS (TE = 30 ms) ^31^P (TE = 2.3 ms) 3T	centrum semiovale (including ACC)	GM: Glx+, NAA+, Cr+, Cho+, lipid catabolites−
Godlewska et al. (2017) [22]	AN	18–41	female	13/12	^1^H SVS STEAM (TE = 11 ms) 7T	ACC, occipital cortex and putamen	Glu− in all areas mI− in ACC and occipital lobe

Abbreviations: ACC = anterior cingulate cortex; AN = anorexia nervosa; BN = bulimia nervose; Cho = choline; re = creatine; dlPFC = dorsolateral prefrontal cortex; Glx = glutamate+glutamine; GM = grey matter; HC = healthy control; mI = myo-inositol; NAA = *N*-acetylaspartate; P total = total phosphate; PDE = phosphodiesters; SVS = single voxel spectroscopy; TE = echo time; WM = white matter; ^1^H = proton spectroscopy; ^31^P = phosphate spectroscopy. * overlapping samples.

**Table 2 jcm-09-01292-t002:** Overview of demographic and psychometric data.

	Anorexia (*n* = 32)		Recovered (*n* = 21)		Healthy Controls (*n* = 33)		ANOVA	Post hoc *t*-Test Tukey-Kramer *
	Mean	SD	Mean	SD	Mean	SD	(d.f.; F; *p*)	
Age (Years)	23.7	4.2	27.6	5.2	23.2	3.4	2, 83; 5.8, 0.004 *	REC > AN, HC
Current BMI (kg/m^2^)	16.2	1.3	20.6	1.3	22.1	2.4	2, 83; 91.0; <0.001 *	HC > REC > AN
Lowest-Lifetime BMI (kg/m^2^)	14.8	1.4	15.1	2.2	-	-	2, 49; 0.5; 0.5	-
Calorie Intake at Breakfast (kcal)	135.9	161.9	297.7	128.1	399.3	86.4	2, 83; 23.7; <0.001 *	HC, RE > AN
EDE Total Score	3.2	1.2	0.6	0.4	0.4	0.3	2, 83; 119.9; <0.001 *	AN > REC, HC
EDE Restraint	3.4	1.4	0.5	0.7	0.4	0.5	2, 79; 90.2; <0.001 *	AN > REC, HC
EDE Eating Concern	2.6	1.5	0.1	0.2	0.0	0.1	2, 79; 76.3; <0.001 *	AN > REC, HC
EDE Weight Concern	3.1	1.4	0.7	0.5	0.4	0.3	2, 79; 76.4; <0.001 *	AN > REC, HC
EDE Shape Concern	3.5	1.4	0.8	0.7	0.6	0.5	2, 79; 88.1, <0.001 *	AN > REC, HC
EDI- Total Score	61.8	9.9	47.2	5.1	44.1	2.8	2, 83; 61.4; <0.001*	AN > REC, HC
Duration of Illness (Years)	6.4	3.9	6.1	5.4	-	-	2, 46; 0.6; 0.5	-
SIAB III	7.4	4.8	4.5	5.5	2.7	3.2	2, 83; 9.1; <0.001 *	AN > REC, HC
BDI-II	21.7	9.7	6.5	6.1	1.6	2.3	2, 83; 75.6; <0.001 *	AN > REC, HC
MWTB	27.9	5.2	29.2	4.8	28.3	4.8	2, 83; 0.5; 0.6	-
Partial GM Volume **	0.59	0.1	0.63	0.1	0.59	0.1	2,83; 2.3, 0.112	-
Partial WM Volume **	0.30	0.1	0.27	0.1	0.32	0.1	2,83; 2.1, 0.134	-
Partial CSF Volume **	0.11	0.0	0.11	0.1	0.10	0.0	2,83; 1.6, 0.202	-

Abbreviation: BMI = body mass index, d.f. = degrees of freedom, EDE = eating disorder examination, EDI = Eating Disorder Inventory–2, SIAB III = Structured Interview for Anorexic and Bulimic Syndromes, Subscale III, BDI-II = Beck Depression Inventory-II, MWT-B = Multiwortwahltest for IQ assessment, AN = anorexia nervosa; HC = healthy control; REC = recovered. * significant group effects, ** partial volume of the left insular MRS voxel.

**Table 3 jcm-09-01292-t003:** Robust one-way ANOVA results for the five metabolites compared across groups and *p*-value adjusted for multiple comparisons using false discovery rate (FDR) correction.

Metabolite	ANOVA	Post-Hoc Lincon
Cho	F (2.32) = 0.53; *p* = 0.593; ξ = 0.17	-
mI	F (2.31) = 1.06; *p* = 0.360; ξ = 0.22	-
NAA	F (2.31) = 3.76; *p* = 0.034 *; ξ = 0.36	AN < HC
Glx	F (2.28) = 0.39; *p* = 0.678; ξ = 0.11	-
Cre	F (2.33) = 0.04; *p* = 0.963; ξ = 0.11	-

Abbreviations: Cho = choline; mI = myo-inositol; NAA = *N*-acetylaspartate; lx = glutamate+glutamine; Cre = creatine. AN = anorexia nervosa; HC = healthy control; ξ = effect size. * significant group effects.

**Table 4 jcm-09-01292-t004:** Robust correlation results for the five metabolites with eating disorder symptom severity according to the eating disorder examination (EDE) scale in the AN group. *p*-values adjusted for multiple comparisons using FDR correction.

Metabolite	Total Score	Restrain	Eating Concern	Weight Concern	Shape Concern
Cho	r = −0.09 *p* = 0.643 *p*_FDR_ = 0.804	r = −0.18 *p* = 0.315 *p*_FDR_ = 0.525	r = −0.09 *p* = 0.636 *p*_FDR_ = 0.655	r = −0.17 *p* = 0.365 *p*_FDR_ = 0.457	r = −0.12 *p* = 0.510 *p*_FDR_ = 0.616
mI	r = 0.03 *p* = 0.868 *p*_FDR_ = 0.868	r = 0.31 *p* = 0.080 *p*_FDR_ = 0.398	r = 0.08 *p* = 0.655 *p*_FDR_ = 0.655	r = −0.06 *p* = 0.733 *p*_FDR_ = 0.733	r = −0.09 *p* = 0.616 *p*_FDR_ = 0.616
NAA	r = −0.28 *p* = 0.120 *p*_FDR_ = 0.300	r = −0.06 *p* = 0.745 *p*_FDR_ = 0.849	r = −0.25 *p* = 0.174 *p*_FDR_ = 0.655	r = −0.46 *p* = 0.008* *p*_FDR_ = 0.020	r = −0.25 *p* = 0.171 *p*_FDR_ = 0.322
Glx	r = −0.33 *p* = 0.061 *p*_FDR_ = 0.300	r = −0.04 *p* = 0.849 *p*_FDR_ = 0.849	r = −0.16 *p* = 0.377 *p*_FDR_ = 0.655	r = −0.48 *p* = 0.005 * *p*_FDR_ = 0.020	r = −0.40 *p* = 0.025 * *p*_FDR_ = 0.126
Cre	r = −0.15 *p* = 0.423 *p*_FDR_ = 0.705	r = −0.23 *p* = 0.215 *p*_FDR_ = 0.525	r = −0.13 *p* = 0.489 *p*_FDR_ = 0.655	r = −0.25 *p* = 0.169 *p*_FDR_ = 0.281	r = −0.23 *p* = 0.193 *p*_FDR_ = 0.322

Abbreviations: Cho = choline; mI = myo-inositol; NAA = *N*-acetylaspartate; Glx = glutamate+glutamine; Cre = creatine. AN = anorexia nervosa; FDR = false discovery rate. * significant correlations.

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
