# Peer review of "Insular Cell Integrity Markers Linked to Weight Concern in Anorexia Nervosa—An MR-Spectroscopy Study"

_jcm, 2020, doi:10.3390/jcm9051292_

Round 1
Reviewer 1 Report
This paper is well written and engaging, setting out the gap in the literature, justifying the approach, presenting the Results clearly and discussing them thoughtfully.
Given that the data are reasonably straight forward and that appropriate controls and corrections (metabolites with respect to partial WM /GM volumes) are in place, the major consideration for this reviewer is in the interpretation of the data.
Despite some effort to assign causality between metabolites such as NAA and Glx that align with the integrity (and possibly functionality) of neurons in the insula there remains some confusion (for me) in this regard. The proposal that developmental dysregulation of the neuronal populations in the insular, possibly in conjunction with adverse life events, might impact concerns about body weight and bodily awareness is an attractive idea that, as recognised by the authors, has been incorporated into models of insular integration that bring together not only perceptions of bodily state but functional parameters including hunger, autonomic control and attitudes. But the problem with this chain of events is that this dysregulation disappears in the REC group consistent with it being a "state" characteristic.The obvious extension of this would be that starvation and emaciation of the AN group has driven (and reinforced) the deficit in the insular that disappears with weight gain; however, the is no correlation with BMI. There is acknowledgement that lower Glu concentrations in AN patients may reflect differences in activity rather than just cell integrity and that this would fit with fMRI evidence but this still does not help with any unifying suggestion as to what is driving what in this situation. It would appear that the data speaks against both brain dysfunction causing aberrant behaviour AND behavior (and its metabolic consequences) causing brain dysfunction.
In an otherwise excellent discussion it would be helpful to try to come to grips with this.
Author Response
We thank both reviewers for their helpful and constructive comments and suggestions. We have revised the manuscript accordingly and answer the open questions of the reviewers in the following
In addition, we have corrected some linguistic and stylistic errors.
Reviewer 1:
Despite some effort to assign causality between metabolites such as NAA and Glx that align with the integrity (and possibly functionality) of neurons in the insula there remains some confusion (for me) in this regard. The proposal that developmental dysregulation of the neuronal populations in the insular, possibly in conjunction with adverse life events, might impact concerns about body weight and bodily awareness is an attractive idea that, as recognized by the authors, has been incorporated into models of insular integration that bring together not only perceptions of bodily state but functional parameters including hunger, autonomic control and attitudes. But the problem with this chain of events is that this dysregulation disappears in the REC group consistent with it being a "state" characteristic. The obvious extension of this would be that starvation and emaciation of the AN group has driven (and reinforced) the deficit in the insular that disappears with weight gain; however, the is no correlation with BMI. There is acknowledgement that lower Glu concentrations in AN patients may reflect differences in activity rather than just cell integrity and that this would fit with fMRI evidence but this still does not help with any unifying suggestion as to what is driving what in this situation. It would appear that the data speaks against both brain dysfunction causing aberrant behavior AND behavior (and its metabolic consequences) causing brain dysfunction.
In an otherwise excellent discussion it would be helpful to try to come to grips with this.
We thank the reviewer for the positive perception of our manuscript including the discussion.
The point raised relates to principles of interpreting scientific data: We completely agree that no simple and straight-forward causality can be derived from the data. We discuss this explicitly on page 14 in the revised version (see also below). Beforehand, we want to answer in some more detail with respect to this issue:
After an association has been described and acknowledged, there is often a temptation to draw "hasty" conclusions regarding causality. To further complicate matters, associations can even represent epiphenomena. With the cross-sectional design of this study we cannot proof that the recruited sample of REC might represent a “non-representative” subgroup of former AN with a good prognosis. They might therefore already have had a normal NAA signal during their acute stage. We briefly mention this aspect on P12L396-8. (We had also discussed this issue with respect to our fMRI data on instructed fear [Maier et al., Psychother Psychosom, DOI: 10.1159/000495367]).
Indeed, correlations with the BMI in the current sample would have made arguments clearly easier. Though one might be reminded that even in the case of structural aberrations in AN, which are largely comparable to healthy control subjects after weight restoration [Nickel et al., Int J Eat Disord. DOI.org/10.1002/eat.22918] some studies failed to show positive correlations of grey matter volume with the BMI [e.g. Fonville et al., Psychological Medicine. DOI:10.1017/S0033291713002389]. This might be due to the limited power as a result of limited samples sizes or large interindividual variance.
In this study, other reasons for the absence of BMI correlations (and associations with weight concern on the other hand) might be related to factors such as hormonal status, osmotic dysregulation and hydration status.
In summary, we agree with the reviewer that the conclusions with respect to causality cannot be drawn and possible explanations should be described cautiously. In the revised version, we therefore discuss causality much more cautiously. We rewrote the first paragraphh of 4.1. (P11L346), the fourth paragraph of 4.2. (P12L405-9) and the summary (P13L451), because the initial discussion might have been too suggestive of a causal relation of brain structure, metabolism and psychometric data. In the sentences (P11L357-64), we had already pointed to various possibly influencing factors, which we are currently not able to disentangle.
On P12L396-403 we now explicitly stress the importance of this topic and finally on P12L411-14 in the context of recovered participants.
Reviewer 2 Report
The paper is about an interesting topic and it is well explained and the research is well designed.
I have some comments:
- A recent consensus paper from Martin et al. (2020) [doi: 10.1002/mrm.27742] suggests to perform both water suppression and shimming, as well as they suggest to use a semi-LASER methodology over a PRESS one for 3 T MRSI. Authors state only about water saturation. Methodolody should be implemented.
- The authors state that this sample is quite the same of another study that found alteration in the insula of the patients. Could this be a bias for the authors? I think this could explain a part of the results and should be taken under deeper discussion.
- Conclusion seems to be too much enthusiastic. A cross sectional study cannot give information about the state or the trait nature of a characteristic of a disorder complex as anorexia nervosa. I think the authors should change that part.
- HC was tested also for previous eating disorders? Because the "lowest BMI" camp is empty and it is not clear.
- Authors state that a standardized breakfast was performed before the scans. There are any differences between AN and HC breakfast? How have the authors checked the real calories assumed?
- Author Contribution is empty.
Author Response
We thank both reviewers for their helpful and constructive comments and suggestions. We have revised the manuscript accordingly and answer the open questions of the reviewers in the following
In addition, we have corrected some linguistic and stylistic errors.
Reviewer 2:
A recent consensus paper from Martin et al. (2020) [doi: 10.1002/mrm.27742] suggests to perform both water suppression and shimming, as well as they suggest to use a semi-LASER methodology over a PRESS one for 3 T MRSI. Authors state only about water saturation. Methodolody should be implemented.
Thank you for suggesting the important consensus paper.
We agree with Martin et al. (2020) that semi-LASER sequence would be preferable. Yet, at the time of study initiation no semi-LASER product sequences were available for Siemens scanners. Later CSI product sequences became available, and we implemented single-voxel semi-Laser sequences later, when the study was already running. Semi-LASER protocols reduce the chemical shift artifacts which occur in PRESS sequences. This artifact leads to a displacement of the metabolite location of approx. 2 mm. Water suppression (saturation) was only disabled for water-reference spectrum. We added the following information about manual shimming adjustments on P6L213: “Shimming parameters were further manually adjusted to minimize the full-width at half maximum of the water peak.”
The authors state that this sample is quite the same of another study that found alteration in the insula of the patients. Could this be a bias for the authors? I think this could explain a part of the results and should be taken under deeper discussion.
This is an important point. However, we had conceptualized the set-up of the whole study beforehand. This means that the location of the MRS ROI was chosen due based on the function reported in AN literature. All data were acquired in the same session. We agree that insular findings within the same sample, even if obtained with different modalities, have to be interpreted with some caution. Nevertheless, converging findings from different modalities support the evidence for an insular involvement in the pathophysiology of AN. We now discuss this aspect in the limitations section P13L437-9: “The overlap of the reported structural, functional, and metabolic alterations in the insular cortex in AN, which was investigated in the same subjects, should be confirmed in an independent sample.”
Conclusion seems to be too much enthusiastic. A cross sectional study cannot give information about the state or the trait nature of a characteristic of a disorder complex as anorexia nervosa. I think the authors should change that part.
Thank you for your important remark. In the revised version of the manuscript we have rephrased the according sentences: P12L405-9 “The absence of the NAA signal reductions in women who had recovered from AN compared to those with AN suggests that the apparent loss of neuronal density/integrity (apart from volume changes, e.g., due to dehydration effects) seem to be of particular importance in acute patients” and P13L450-3: ”These data also improve our understanding of the pathophysiology of acute and recovered patients with AN, which can influence both our clinical thinking and psychoeducational interventions [84].” Furthermore, we have been more cautious with any causal statements (see Reviewer #1).
HC was tested also for previous eating disorders? Because the "lowest BMI" camp is empty and it is not clear.
We assessed previous eating disorder symptomatology using the SKID-I. Any earlier eating disorder symptoms were defined as exclusion criteria for the control sample.
Lowest lifetime BMI was not assessed in the HC group, since many participants were unaware of their lowest body weight.
Authors state that a standardized breakfast was performed before the scans. There are any differences between AN and HC breakfast? How have the authors checked the real calories assumed?
The breakfast always consisted of the same components. Therefore, we were able to assess calories digested. Yet, as expected, AN consumed less calories than HC, which we now list in table 2. However, AN patients consume fewer calories due to illness, which therefore cannot be harmonized across groups under ethical aspects. Yet, it is rather unlikely that a single "standard breakfast" will completely normalize all metabolic anomalies we detected in the MR spectroscopy signals.
Author Contribution is empty.
We are not sure why the Author contribution is not visible to you. Below you find the list of contribution:
Conceptualization, Simon Maier, Evgeniy Perlov, Almut Zeeck, Ludger Tebartz van Elst and Andreas Joos; Data curation, Simon Maier, Alina Kukies, Lukas Holovics and Michael Dacko; Formal analysis, Armin Hartmann; Funding acquisition, Almut Zeeck and Andreas Joos; Investigation, Simon Maier, Kathrin Nickel and Alina Kukies; Methodology, Simon Maier, Evgeniy Perlov, Dominique Endres, Lukas Holovics, Armin Hartmann, Michael Dacko and Thomas Lange; Project administration, Simon Maier and Andreas Joos; Resources, Andreas Joos; Software, Michael Dacko and Thomas Lange; Visualization, Simon Maier; Writing – original draft, Simon Maier and Alina Kukies; Writing – review & editing, Kathrin Nickel, Almut Zeeck, Ludger Tebartz van Elst, Dominique Endres, Derek Spieler and Andreas Joos.
Round 2
Reviewer 2 Report
I am glad that my suggestions have helped the authors. I believe that in the present form the article can be accepted.